# Influence Mechanism of Ultrasonic Vibration Substrate on Strengthening the Mechanical Properties of Fused Deposition Modeling

**DOI:** 10.3390/polym14050904

**Published:** 2022-02-24

**Authors:** Wenzheng Wu, Jialin Li, Jili Jiang, Qingping Liu, Aodu Zheng, Zheng Zhang, Ji Zhao, Luquan Ren, Guiwei Li

**Affiliations:** 1Advanced Materials Additive Manufacturing ((AM)2) Lab, School of Mechanical and Aerospace Engineering, Jilin University, Changchun 130025, China; wzwu@jlu.edu.cn (W.W.); lijl9919@mails.jlu.edu.cn (J.L.); jiangjl16@mails.jlu.edu.cn (J.J.); zhengad21@mails.jlu.edu.cn (A.Z.); zhengzhang21@mails.jlu.edu.cn (Z.Z.); jzhao@jlu.edu.cn (J.Z.); lqren@jlu.edu.cn (L.R.); 2Chongqing Research Institute, Jilin University, Longxing Town, Yubei District, Chongqing 401123, China; 3Key Laboratory of Bionic Engineering (Ministry of Education), Jilin University, Changchun 130022, China; liuqp@jlu.edu.cn

**Keywords:** fused deposition modeling, ultrasonic-assisted, 3D printing, mechanical property

## Abstract

Fused deposition modeling is the most widely used 3D-printing technology, with the advantage of being an accessible forming process. However, the poor mechanical properties of the formed parts limit its application in engineering. Herein, a new ultrasonic-assisted fused deposition modeling 3D-printing method was proposed to improve the mechanical properties of the formed parts. The effects of ultrasonic vibration substrate process parameters and printing process parameters on the tensile and bending properties of formed samples were studied. The tensile strength and bending strength of the samples printed with a 12 μm ultrasonic amplitude can be increased by 13.2% and 12.6%, respectively, compared with those printed without ultrasonic vibration. The influence mechanism of ultrasonic vibration on mechanical properties was studied through microscopic characterization and in situ infrared monitoring experiments. During the printing process, increasing the ultrasonic vibration and temperature employed via the ultrasonic substrate can reduce the pore defects inside the sample. The mechanical properties of FDM-formed samples can be controlled by adjusting ultrasonic-assisted process parameters, which can broaden the application of 3D printing.

## 1. Introduction

Fused deposition modeling (FDM) technology has the advantages of low equipment cost, low maintenance cost, low material cost, high material utilization efficiency, a wide variety of materials, and low environmental pollution, and has been widely used in the fields of education, the automobile industry, biomedical engineering, construction, and aerospace [1,2,3,4,5]. Although the application prospects of FDM technology are very broad, there is still a certain gap between the mechanical properties of the formed parts and traditional manufacturing methods [6]. Due to the characteristics of the FDM deposition manufacturing process and its associated defects, such as pores in the internal wires of the sample parts, the popularization and application of FDM technology in various fields are greatly limited [7,8]. 

Recently, scholars have conducted many relevant research efforts to improve mechanical properties via the adjusting of printing process parameters and composite material modification [9,10,11]. Wenli Ye et al. explored the influence mechanism of continuous fiber on PPS-composite-forming samples [12,13]. Lederle Felix et al. studied the effect of a nitrogen-protected atmosphere on the mechanical properties of an ABS sample and a nylon sample: the tensile strength of the ABS sample increased by about 10%, and that of the nylon sample increased by about 30% [14]. Durgun Ismail et al. studied the influence of the construction direction and filling angle on the mechanical properties of FDM-formed parts by taking ABS material as the research object and concluded that the influence of the fused deposition modeling construction direction on mechanical properties is more significant than that of the filling angle [15]. Feng Zhao et al. took polyether ether ketone (PEEK) material as the research object to study the influence of forming environment temperature, nozzle temperature, forming substrate temperature, and other parameters on the tensile strength of the formed sample [16,17]. The experiment showed that reasonable temperature control could effectively improve the bonding force between two adjacent rasters. At the same time, the influence of ambient temperature, nozzle temperature, and forming substrate temperature on the tensile strength of the forming sample was obtained from low to high [18,19,20]. Xinhua Liu et al. studied the influence of process parameters such as printing direction, printing layer thickness, printing line width, and filling density on the warpage deformation, surface roughness, and mechanical properties of formed parts [21]. Yande Liang et al. combined theoretical and experimental analyses to summarize and analyze the influence of heat dissipation conditions on the bond strength and warpage deformation between rasters of formed parts in the forming process [22].

Gunduz et al. conducted relevant research on the ultrasonic-vibration-assisted 3D-printing of high-viscosity materials, introduced high-amplitude ultrasonic vibration into the nozzle, and finally obtained commercial polymer clay with a viscosity up to 14,000 pa·s [23]. Richard et al. found that ultrasonic treatment can effectively improve the surface finish, reduce the porosity under the surface, increase the surface hardness, and improve the fatigue performance of 3D-printed metal parts [24,25]. Alireza et al. explored the effect of an ultrasonic vibration print head on the bonding interface strength of printing wire produced by FDM [26]. When ultrasonic vibration was added to the process of printing wire, the interlayer bonding strength increased by up to 10%.

Guiwei Li et al. demonstrated that the ultrasonic postprocessing of 3D-printed samples resulted in the refusion of defects within the samples [27,28]. In this paper, based on the existing FDM technology, an ultrasonic-vibration-assisted method to enhance the mechanical properties of FDM additive manufacturing parts is proposed. An ultrasonic-vibration-assisted FDM additive manufacturing printer was developed, and the effects of ultrasonic process parameters and printing process parameters on the tensile and bending properties of formed samples were studied. The influence mechanism of ultrasonic vibration on mechanical properties was studied by microscopic characterization, differential scanning calorimetry, and in situ infrared monitoring experiments.

## 2. Materials and Methods

In this paper, an ultrasonic-vibration-assisted FDM additive manufacturing device is developed. The whole device is mainly composed of an ultrasonic vibration platform system and a 3D-printing system. A schematic diagram of the device is shown in Figure 1.

During the processing, the substrate of the ultrasonic vibration system continues to produce high-frequency mechanical vibration, and the thermoplastic molten wire is extruded from the printing nozzle and pressed by the nozzle on the substrate or on the deposited rasters, which are printed on the upper layer. The extruding raster and the deposited rasters are driven by the substrate to carry out high-frequency vibration. The temperature at the bonding interface increases, the kinetic ability of the polymer molecular chain is improved, and the diffusion as well as the entanglement speed of the molecular chain at the bonding interface of adjacent wires are promoted, so the bonding strength of the bonding interface is increased, and the mechanical properties of the whole forming sample are improved.

### 2.1. Experiment on the Influence of Ultrasonic Parameters on FDM Samples

The test parts were printed by a homemade ultrasonic-vibration-assisted FDM additive manufacturing device. The 3D-printing material was PLA (Hangzhou Xianlin 3D Science and Technology Co., Ltd., Hangzhou, China). The tensile test sample is designed by a nonstandard design. The shape of the sample was designed as a strip, and the size of the sample was 70 mm × 8 mm × 3 mm. The shape of the bent sample was designed by ISO 178:2001, and the size of the sample was 60 mm × 10 mm × 3 mm.

Based on the self-built ultrasonic-vibration-assisted FDM printer, the ultrasonic amplitude, printing layer thickness, and printing speed were selected as the research objects. The effects of these parameters on the tensile and bending properties of printed samples were investigated. For each group of the test printed samples, the printing temperature was 205 °C, the filling gap was 0 mm, the filling line angle was +45°/−45°, and the number of outer layers was 3. In test group A, the ultrasonic amplitude was selected as the variable, ultrasonic vibration stopped after the eighth layer of the sample printing, and the 3D printer continued to print the remaining parts. In test group B, printing speed was selected as the variable, ultrasonic vibration stopped after the eighth layer of the sample printing, and the 3D printer continued to print the remaining parts. Meanwhile, group B included control tests without ultrasonic vibration reinforcement. In test group C, the printing layer thickness was selected as the variable. The sample with a thickness of 0.15 mm stopped ultrasonic vibration after printing the thirteenth layer, the sample with a thickness of 0.2 mm stopped ultrasonic vibration after printing the tenth layer, and the sample with a thickness of 0.25 mm stopped ultrasonic vibration after printing the eighth layer. The sample with a printing layer thickness of 0.3 mm stops ultrasonic vibration after printing the seventh layer, and the 3D printer continues to print the remaining samples when the ultrasonic vibration stops. Meanwhile, group C contained control tests without ultrasonic vibration reinforcement. To ensure the accuracy of the test, 5 samples were strengthened for each group of test parameters, and the test results were averaged. Table 1 shows the arrangement of the printing test parameters for each group.

### 2.2. Experiment on the Thermal Influence of Ultrasonic-Assisted FDM

This section mainly studies the effects of the surface temperature of printed samples on the process of ultrasonic strengthening; the printed samples were prepared for simulating temperature change in the process of ultrasonic strengthening with a heated substrate later. The ultrasonic vibration stops after the eighth layer of the sample is finished, and the sample continues to form. In test group B, printing speed was selected as the variable, ultrasonic vibration stopped after the eighth layer of sample printing, and the 3D printer continued to print the remaining samples. In test group C, the printing layer thickness was selected as the variable. The sample with a thickness of 0.15 mm stopped ultrasonic vibration after printing the thirteenth layer, the sample with a thickness of 0.2 mm stopped ultrasonic vibration after printing the tenth layer, and the sample with a thickness of 0.25 mm stopped ultrasonic vibration after printing the eighth layer. The sample with a printing layer thickness of 0.3 mm stops ultrasonic vibration after printing the seventh layer. When the ultrasonic vibration stops, the 3D printer continues to print the remaining samples. Other printing parameters: a printing temperature of 205 °C, a filling angle of +45°/−45°, several outer layers (3), and a filling gap of 0 mm. Table 2 shows the arrangement of the printing test parameters for each group. In the 3D-printing process, an infrared thermal imager is used to monitor in situ the changes in the surface temperature of samples during the whole printing process under different parameters. All data points in the experiment are the temperature of the surface after printing to the next layer.

### 2.3. Effect of Temperature on the Mechanical Properties of Samples during Ultrasonic Strengthening

The ultrasonic vibration platform system was replaced with a heating substrate system. The heating substrate was used to simulate the surface temperature change in samples in the ultrasonic vibration printing process, and the mechanical samples were printed to test the variation rule of their mechanical properties. The thermal substrate device is mainly composed of a 24 V heating substrate, a 24 V AC-to-DC power supply, a K-type thermocouple, and a REX-C100 electronic temperature controller. The 24 V heating substrate creates a temperature range of room temperature to 200 °C.

The effect of heat generated in the ultrasonic vibration printing process on the mechanical properties of samples was studied. According to different ultrasonic amplitudes measured and different 3D-printing parameters with ultrasonic amplitudes of 12 μm, the mechanical samples were printed by the heating substrate to simulate the temperature changes in the process of ultrasonic vibration printing.

The sample is formed by a 3D printer installed with a heated substrate. During the printing process, the temperature of the heated substrate is adjusted in real-time according to the temperature curve measured by ultrasonic vibration under the same 3D-printing parameters, so that the temperature of the printed sample in the layer is the same as the surface temperature of ultrasonic vibration printing.

Group A used a heated substrate to simulate the change in surface temperature during the printing process of tensile samples and bending samples under different ultrasonic amplitudes, printed tensile samples, and bending samples. The actual ultrasonic amplitude was 0 μm. Group B used a heated substrate to simulate the temperature change in sample surface during printing under a fixed ultrasonic amplitude and different printing speeds, printed tensile samples, and bending samples; at the same time, experimental group B was added to the control group without simulated temperature heat treatment, and other experimental parameters were the same as those of experimental group B. The actual ultrasonic amplitude of the two groups was 0 μm. In experimental group C, the surface temperature changes in samples during printing, tensile samples, and bending samples were simulated with a heated substrate under a fixed ultrasonic amplitude and different printing layer thicknesses; at the same time, group C was added to the control test group without simulated temperature heat treatment, and other experimental parameters were the same as those of experimental group C. The actual ultrasonic amplitude of the two groups was 0 μm. Table 3 shows the arrangement of printing test parameters for each group.

### 2.4. Characterization Test Equipment

The tensile properties and bending properties of the samples were tested by an UTM6104 electronic universal testing machine (Shenzhen Sansi Longitudinal and Horizontal Technology Co., Ltd., Shenzhen, China). The glass transition temperature, Tg, cold crystallization peak temperature, Tc, melting peak temperature, Tm, and crystallinity of the samples were analyzed by differential scanning calorimetry (DSC). The test equipment was Q2000 differential scanning calorimetry (DSC) (New Castle, DE, USA). A DSC test using a nitrogen atmosphere, a temperature rise rate of 10 °C/min, and a temperature range of 25 °C to 200 °C selected different parts of the same position of the test material. DSC detection adopts a nitrogen protective atmosphere, a temperature rise rate of 10 °C/min, a temperature measurement range of 25 °C to 200 °C and selects different parts of the same position of the test material. During the thermal conductivity test, the sample surface temperature was collected by a Flira 310 thermal imager (FLIR Systems, Inc., Wilsonville, OR, USA), with a detection accuracy of ±2 °C and a temperature range of 0 to 350 °C. The heating block temperature control device in the process of material thermal conductivity detection adopts an AT80017 electronic temperature controller (Shenzhen Arthur Technology Co., LTD., Shenzhen, China). The heating block adopts a 220 V aluminum heating block.

## 3. Discussion and Analysis

### 3.1. Influence of Ultrasonic Parameters on Mechanical Properties 

#### 3.1.1. Influence of Ultrasonic Amplitude on Mechanical Properties

Figure 2 shows DSC curves of samples under different ultrasonic amplitudes. Samples are all from the same sections of samples under different amplitudes, so as to avoid errors of test results caused by differences in different positions. According to the DSC curves, the crystallinity of samples under different ultrasonic amplitudes can be calculated as $X_{c}$; the calculation formula is as follows (Equation (1)): *Xc* = [(Δ*H_m_* + Δ*H_c_*)/Δ*H_theory_*] × 100%(1)
where $X_{c}$ is the crystallinity, $△H_{m}$ is the enthalpy of melting, $△H_{c}$ is the enthalpy of cold crystallization, $△H_{Theory}$ is the theoretical melting enthalpy of 100% crystallization of the tested sample, and the theoretical melting enthalpy of PLA is 93.7 J/g [28].

It can be seen from Figure 2 that the Tg of the glass-state transition temperature of samples under different ultrasonic amplitudes is within the range of 59.0 ± 0.3 °C, with little change. The melting peak temperature, Tm, is 163.8 ± 0.3 °C, and the peak height and peak width of the melting peak have no significant change. The temperature, Tc, of the cold crystallization peak decreases first and then increases with an increase in ultrasonic amplitude. At an ultrasonic amplitude of 0 μm, that is, without ultrasonic vibration, the Tc of a sample is 105 °C, and the maximum Tc range is 3 °C with ultrasonic vibration. The crystallinity of the samples without ultrasonic vibration is less than that after ultrasonic vibration, and the crystallinity of the samples with a different ultrasonic amplitude is almost the same. The crystallinity of the sample with ultrasonic vibration increases because intramolecular friction heat is generated in the printed sample during the process of ultrasonic vibration, and a certain friction heat is generated between the surface of the deposited rasters and the deposited layers. Within a period of time, the temperature at the bonding interface of the sample is higher than that without ultrasonic vibration, and the heat dissipation conditions are the same. Therefore, the crystallinity of the sample with ultrasonic vibration increases.

Figure 3a,b show the curves of the tensile strength and flexural properties of printed samples with different ultrasonic amplitudes. It can be seen that, with an increase in ultrasonic amplitude, the bending strength of the 3D-printed sample increases first and then slightly decreases, and that the tensile strength and bending modulus both increase gradually. When the ultrasonic amplitude is 13 μm, the tensile strength and bending modulus of the 3D-printed sample reach their maximum values, which are 41.37 MPa and 2734.74 MPa, respectively, increases of 13.1 percent and 14.9 percent. When the ultrasonic amplitude is 12 μm, the bending strength of the 3D-printed sample reaches its maximum value, which is 81.45 MPa, 12.6% higher than that of the sample with 0 μm ultrasonic vibration.

As can be seen in Figure 4a, without ultrasonic vibration the fracture surfaces of tensile samples showed their raster lap together, and the samples are within the same layer between two adjacent wires; adjacent between two layers of wire is a small bonding area, obviously a wire contour, so the tensile samples may start with some single-wire fracture; therefore, the overall tensile property of the sample is low. As can be seen in Figure 4b, a sample-packed bed bottom area is sticky on the surface of the junction between several layers of wire in the 0 μm ultrasonic vibration sample underside of the bonding area; adjacent wires between the fusion effect are better, but some upward filling layer between the wire and the wire-bonding effect are not obvious improvements. Outside the outline layer, the wire junction plane between the bonding area increased. This shows that when ultrasonic vibration is low, the diffusion-promotion effect of the polymer molecular chain between the deposition wire and deposition layer is small; in Figure 4c, the fusion between the filaments of the sample filling layer is improved to some extent compared with Figure 4a, but the improvement effect is still not particularly obvious. The bonding area between the filaments of the sample contour layer is better than that of the filaments in Figure 4a, and the pores between the contour layers are smaller; as shown in Figure 4d, the bonding area between the internal filling layers and the adjacent wires has been greatly improved. In some areas, the bonding interface of adjacent wires have achieved a good fusion, with no obvious silk contour and interface visible; the filling gap has also become significantly smaller. Through the analysis of the above four cross-sections, it is shown that when the ultrasonic vibration is large there is more internal friction heat between molecules and friction heat generated between the deposition layer silk and the deposition layer at the same time, and that there is greater energy given to the molecular chain by the ultrasonic vibration, so that the macromolecular chain and chain segment can crosslink and diffuse faster between the bonding interface layers of adjacent silk during the deposition process. Furthermore, the bonding strength of the bonding interface is increased, so as to improve the load that can be borne per unit area. 

According to the analysis of the mechanical properties, DSC curve, and section morphology, the larger the ultrasonic amplitude the larger the bonding area between adjacent wires in the sample, the smaller the filling gap, the better the fusion effect at the bonding interface between adjacent wires, and the greater the load-bearing capacity per unit area. At the same time, the addition of ultrasonic vibration can reduce the internal stress and improve the crystallinity of the sample in the printing process. An increase in crystallinity causes the internal macromolecular chains to be more closely arranged and increases the ability to resist load per unit area. Therefore, with an increase in ultrasonic amplitude, the tensile strength and bending performance of the sample increase.

#### 3.1.2. Influence of Printing Speed and Ultrasonic Amplitude on the Mechanical Properties of Samples

Figure 5a,b show the variation curves of the tensile strength, bending strength, and modulus of 3D-printed samples with and without ultrasonic vibration at different printing speeds with an ultrasonic amplitude of 12 μm. As can be seen from Figure 5a,b, when ultrasonic vibration is not added in the forming process of 3D-printed samples the tensile strength, bending strength, and bending modulus of printed samples gradually decrease with an increase in printing speed. The main reason is that when the printing speed is low and the deposition layer is deposited to the next layer, the surface temperature of the last printed deposition layer is relatively low, which is not conducive to the diffusion of the polymer chain between the bonding interface between the deposited wire and the previous deposited wire. However, the nozzle stays at the current deposition position for a long time and applies pressure, which can continuously heat transfer and pressurize the deposited wire, the surface temperature at the bonding interface between the deposited wire and the adjacent wire is kept high for a long time, which increases the bonding interface area and softening as well as wetting time, enables the polymer molecular chain at the bonding interface to diffuse and wind well, and improves the bonding strength of the bonding interface. At the same time, excessive speed easily leads to an uneven wire diameter; under the interaction of the three, the favorable influence is greater than the adverse influence. Therefore, the smaller the printing speed in a certain range the higher the tensile strength, bending strength, and bending modulus of the formed sample.

When the ultrasonic amplitude of 12 μm was added in the forming process, the variation trends of the tensile strength, bending strength, and bending modulus of the sample were similar to that of the sample without ultrasonic vibration with the increase of printing speed. The tensile strength, flexural strength, and flexural modulus of 3D-printed samples with a 12 μm ultrasonic amplitude increased by 8.5%, 10.1%, and 7.2%, and the maximum values increased by 13.2%, 12.6%, and 12.6%, respectively, compared with those without ultrasonic vibration at the same printing speed. With a greater ultrasonic vibration sample strength and modulus than those under the same printing speed without ultrasonic vibration strength and high modulus of samples, and for ultrasonic vibration to further promote the sedimentary wire and adjacent wire combination of the contact area of the interface, the interface temperature and molecular chain of sports ability cause the combination of the interface between the polymer molecular chain and chain segment diffusion and winding speed. Therefore, the properties at the interface are closer to the internal properties of the single wire, and the mechanical properties of the sample are further improved. It can be seen from Figure 5a,b that the tensile strength, bending strength, and bending modulus of the printed sample with a speed of 60 mm/s plus a 12 μm ultrasonic amplitude is higher than that of the sample without ultrasonic reinforcement at 40 mm/s. Therefore, the application of ultrasonic-vibration-assisted melt deposition additive manufacturing technology in practical engineering can improve the printing speed without sacrificing strength, as well as improve production efficiency.

#### 3.1.3. The Effects of Layer Thickness and Ultrasonic Amplitude on Mechanical Properties

Figure 6a,b show the curves of the tensile strength, flexural strength, and modulus of 3D-printed samples with an ultrasonic amplitude of 12 μm and without ultrasonic vibration. By observing Figure 6a,b, it can be seen that the tensile strength, bending strength, and bending modulus of the printed sample without ultrasonic vibration gradually decrease with an increase in the printing layer thickness. The tensile strength, flexural strength, and modulus of the sample printed with a 12 μm ultrasonic amplitude also decrease with an increase in the printing layer thickness. The tensile strength, bending strength, and bending modulus of the printed samples with ultrasonic vibration increased by 8.8%, 8.0%, and 4.6%, and the maximum values increased by 18.0%, 12.6%, and 14.2%, respectively, compared with those without ultrasonic vibration at the same printing layer thickness. The tensile strength, flexural strength, and modulus of the sample printed with 0.25 mm thickness and a 12 μm ultrasonic amplitude are higher than those of the sample printed with 0.15 mm thickness and without ultrasonic vibration, which can greatly shorten the printing time and meet the strength requirements when the surface topography of the product is not highly required in practical applications.

Figure 7 shows the section morphologies of the printed sample without ultrasonic vibration in addition to the printed sample with an ultrasonic amplitude of 12 μm at 0.2 mm and 0.3 mm layers. It can be observed from Figure 7a,c that, in the 3D-printing forming process, the smaller the printing thickness of each layer is, the smaller the gap between the filaments of the filling layer inside the sample and the larger the binding area between the filaments of the filling layer. This is because, when printing samples with different layer thicknesses, the diameter of the printer nozzle is the same, as is the diameter of the extruded wire. When the thickness of the printing layer is small, the height of the nozzle to rise after printing each layer is correspondingly small. During the printing process, the nozzle exerts greater pressure on the freshly extruded wire, so that the freshly extruded wire is extruded into a corresponding smaller layer thickness, increasing the contact area between the newly extruded wire; the upper layer of deposited wire increases the diffusion and entanglement of the polymer chain between the joint surfaces of the two layers of deposited wire. At the same time, the increase in pressure also increases the diffusion of the polymer chain at the bonding interface, and finally enhances the tensile strength and bending strength of the 3D-printed sample; when the thickness of the printing layer is large, the height of each layer to be raised after printing by the nozzle is large, and the extrusion force of the corresponding nozzle on the wire of the just extruded nozzle is small, so that the contact area and bonding force between the deposited wires are relatively small. During the strength test, it is easy to lead to the first fracture of a single wire, resulting in the relatively low tensile and bending properties of the sample. From Figure 7a,b, although without ultrasonic vibration under 0.15 mm-thick samples between the internal wire bonding area of relatively large, thick samples had a very big promotion, it can be observed that in many areas, or the combination of the interface between adjacent wire contours, there is no good fusion between the two wires joined 12 μm after ultrasonic vibration. In many areas of the sample filling layer, the contours of the interface between adjacent wires have been invisible. The contact interface has been well-fused, so that the performance of the interface is close to the interior of the wire and the force per unit area can be further increased. It can be observed from Figure 7d that, compared with the sample printed with 0.3 mm thickness without ultrasonic vibration, the area of the bonding interface between the filled wires in the sample with ultrasonic vibration is greatly improved. In many areas, the contour of the bonding interface between adjacent wires is not obvious, and even the contour of a single wire cannot be observed, increasing the force per unit area.

Comprehensive mechanical properties and section morphology analysis show that under the same 3D-printing parameters and 12 μm-ultrasonic-amplitude-strengthened sample tensile strength, the bending strength and modulus are higher than those in the print sample without ultrasonic strengthening, because the print ultrasonic vibration to join in the process of improving the deposition in the process of printing in combination with the interface area, temperature, and the combination of molecular chain movement ability; therefore, the crosslinking, diffusion, and entanglement of polymer molecular chains at the bonding interface are accelerated, the bonding strength and filling gap at the bonding interface are improved, and the mechanical properties of the formed parts are improved. At the same time, according to the research the crystallinity of the sample may also be improved in the process of ultrasonic vibration, such that the arrangement of molecular chains is closer and the stress capacity per unit area of the sample is increased.

### 3.2. Thermal Influence of Ultrasonic-Assisted FDM Process

#### 3.2.1. The Effects of Surface Temperature under Different Ultrasonic Amplitudes

Figure 8a,b show the surface temperature change curves of stretched and bent samples printed to different layers in the 3D-printing process with different ultrasonic amplitudes, measured by a thermal imager. It can be seen from the two figures that the surface temperature of the stretching sample and the bending sample without ultrasonic strengthening increases gradually with an increase in the number of printed layers. When the increase reaches a certain temperature, the surface temperature of the sample almost does not change with an increase in the number of printed layers. Along with the increase in the layer number of printing, the sample surface temperature increase gradually, because along with the increase in the layer number of printing the pattern layers are printed out between each other to keep a certain temperature, the area is small and in contact with the air cooling slowly, making the temperature accumulation increase gradually; as the printing layer increased, the heat input and heat of the just-finished printing layer reaches a certain balance, and the surface temperature of the sample remains in equilibrium. The surface temperature of the stretched sample without ultrasonic amplitude is higher than that of the bent sample without ultrasonic amplitude. Because all 3D-printing parameters of the stretched sample are the same, the printing time of each layer of the bent sample is relatively more, so the surface heat dissipation of the sample is higher after printing one layer.

According to Figure 8a,b, the surface temperature of the sample strengthened with ultrasonic vibration increases gradually with an increase in the number of printing layers. The surface temperature of the sample decreases gradually when the ultrasonic vibration stops at the eighth layer after printing. The surface temperature of the sample with ultrasonic vibration is higher than that of the sample without ultrasonic vibration, because ultrasonic vibration produces friction vibration heat and internal intermolecular friction heat on the deposited printing contact surface. The larger the ultrasonic amplitude is the faster the surface temperature of the sample increases in the printing process, possibly because the larger the ultrasonic amplitude is the more intense the intermolecular friction in the printing process and the more intense the surface friction of the wire being deposited, as well as the greater the heat generated per unit time.

#### 3.2.2. Sample Surface Temperature Changes at Different Printing Speeds

Figure 9a shows the curve of the surface temperature of the stretched sample with the number of printed layers at different 3D-printing speeds with or without a 12 μm ultrasonic amplitude. Figure 9b shows the curve of the surface temperature of the bent sample with the number of printed layers at different 3D-printing speeds of a 12 μm ultrasonic amplitude and no ultrasonic vibration. It can be seen from the two figures that the surface temperature of the sample without ultrasonic vibration increases with the number of printing layers and then tends to be stable. When the sample printed without ultrasonic vibration is printed to the same number of layers, the higher the printing speed the higher the surface temperature of the sample is, because the higher the speed the shorter the heat dissipation time and the less heat dissipation after printing a layer. Under the same printing parameters, when the ultrasonic amplitude of 12 μm was added the surface temperatures of both stretched and bent samples increased gradually with an increase in the number of printing layers. 

#### 3.2.3. Changes in the Surface Temperature of Samples under Different Printing Layer Thicknesses

Figure 10a shows the variation curve of the surface temperature of the stretched sample with the number of printed layers when the ultrasonic amplitude is 12 μm and the ultrasonic vibration is not added. Figure 10b shows the curve of the surface temperature of the bent sample with the number of printed layers when the ultrasonic amplitude is 12 μm and the ultrasonic vibration is not added. It can be seen from the two figures that the surface temperature of the tensile sample and the bending sample printed without ultrasonic vibration increases gradually with an increase in the number of printing layers, and the surface temperature of the sample tends to be stable after reaching a certain temperature. When printing to the same number of layers, the surface temperature of the sample with a smaller layer thickness is lower than that of the sample with a larger layer thickness. Although it takes a short time to complete the printing of a layer with a smaller layer thickness, the surface area of the silk surface in contact with air is large, which increases the surface heat dissipation rate, so the surface temperature of the sample with the same number of layers is lower. Under the same 3D-printing parameters, when printing to the same number of layers the surface temperature of the samples with ultrasonic vibration is higher than that of the printed samples without ultrasonic vibration. Because ultrasonic vibration produces certain friction heat and internal friction heat between molecules on the surface of the deposited silk material, the surface temperature of the samples with ultrasonic vibration is higher.

### 3.3. The Effects of Temperature on the Mechanical Properties of Samples

#### 3.3.1. Influence of Ultrasonic Amplitude and Temperature on the Mechanical Properties of Printed Samples

Figure 11 shows the DSC test curves of printed samples using a heated substrate to simulate the surface temperature changes in samples under different ultrasonic amplitudes. The selected test samples are all in the same position inside the sample, to avoid certain influences of different positions on the test results. It can be seen from the DSC curves that the glass-state transition temperature (Tg) and melting peak temperature (Tm) of the samples treated by heating are 59.0 ± 0.6 °C and 161.3 ± 0.5 °C, respectively. The size of the melting peak does not change significantly. The cold crystallization temperature (Tc) of the samples decreases first and then increases with an increase in the simulated amplitude. Simulated at a 10 μm amplitude, the crystallinity of print samples compared with those without a set of simulated temperatures showed no obvious change, because the heating substrate simulating a 10 μm ultrasonic vibration surface temperature changes in the process of printing sample heating substrates simulating a heating curve of the maximum temperature less than the temperature of cold crystallization start. Samples are not produced through cold crystallization, so did not significantly change crystallinity samples; when simulating the temperature of 11 μm, 12 μm, and 13 μm ultrasonic amplitude samples, the maximum temperature simulated by heating the substrate is greater than the temperature at which the material begins to cold crystallize, so the sample crystallizes during a period higher than the cold crystallization temperature, which increases the crystallinity of the sample.

Figure 12 shows the surface temperature changes in printed samples under different ultrasonic amplitudes simulated by heating the substrate in addition to the section morphology of printed bent samples after mechanical experiments. As can be seen from Figure 12, the bending strength and modulus of the printed sample with a simulated temperature first decrease and then increase with an increase in the simulated amplitude. The bending strength and modulus are both lower than that of the simulated sample with a 0 μm amplitude, and their maximum values are 9.1% and 11.0% lower, respectively. It can be seen from Figure 13a that when no simulated temperature is added, each layer of the sample fracture surface can see an obvious wire interface contour; the contact area between adjacent wires is very small. As can be seen from Figure 13b–d, there is no significant difference in the bonding area between the first layer and the fourth layer of wire materials with the addition of a simulated temperature compared with that without the addition of a simulated temperature; from the fourth layer to the ninth layer, the bonding area between the internal wires of the samples increased much more than that of the group without a simulated temperature, and with the increase in simulated amplitude the bonding area between the wires of these layers increased. As a simulation of the ultrasound amplitude increases from 9 to 10 s, the wire-bonding effect between them worsens, because for print layers between them the higher the temperature of the simulation result in the wire-bonding process the greater the liquidity causing wires to sink, the more a layer is the distance of the wire, and the greater the deposit after the wire-bonding effect worsens.

Based on the above analysis, the bending strength and modulus of the sample with a simulated temperature are both lower than those without a simulated temperature, because during the bending test the higher the middle position is the greater the compressive stress is, and the lower the middle position is the greater the tensile stress is. When the simulated temperature is added, although the bonding of the middle layers is better, the compressive stress and tensile stress are improved, but the tensile stress is small near the center itself and the improvement effect is limited. The upper layer bears large compressive stress because the bonding effect is poor; the compressive stress can be greatly reduced, resulting in the bending strength being lower than that of the group without simulated temperature.

The temperature changes in samples under ultrasonic amplitudes of 0 μm, 10 μm, 11 μm, 12 μm, and 13 μm were simulated by the heating substrate. The tensile strength of printed samples measured is shown in Figure 14. As can be seen from the figure, the tensile strength of the sample increases gradually with an increase in the simulated amplitude, reaching a maximum of 40.1 MPa, 9.3% higher than that of the sample without a simulated temperature. With an increase in the simulated ultrasonic amplitude, the bonding effect between the wires in the stretched sample should be similar to that in the bent sample, because the heat treatment temperature of the two is not very different. The reason for the increase in the tensile strength is that during the tensile test the surface of the whole sample is subjected to tensile stress. The increase in the bonding area and bonding strength of the middle layer can greatly improve the ability to withstand tensile strength. At the same time, the processing temperature is higher than the initial cold crystallization temperature, which improves the crystallinity, causes the entanglement of the molecular chains between the silk materials to be closer in the tensile direction, and also causes the tensile load that can be borne to be larger, so the tensile strength of the final sample is improved.

The bending strength and modulus of the samples with ultrasonic vibration are higher than those without ultrasonic vibration, which are different from the changes in the bending strength and modulus simulated by temperature. This is probably because ultrasonic vibration, only to improve the surface temperature of the deposit wire without raising the sample temperature, make each layer’s sedimentary wire not sink, hit on the performance of the top fashion style without ultrasonic-strengthened samples, and ultrasonic vibration that the movement of the molecular chain ability is stronger, makes the same temperature at the interface between combining spread faster; therefore, the bending strength did not weaken, but increased relative to the sample without ultrasonic strengthening. Using a heated substrate to simulate the temperature of printed samples with ultrasonic amplitudes of 10 μm and 11 μm, the tensile strength of printed samples is higher than that of printed samples with ultrasonic amplitudes of 10 μm and 11 μm. Because ultrasonic amplitudes only increase the surface temperature, the bonding area between internal wires is smaller than that of the group with ultrasonic amplitudes of 10 μm and 11 μm. Moreover, weak ultrasonic vibration enhances the kinetic ability of the molecular chain to a lesser degree, so the tensile strength of the group with ultrasonic vibration is slightly weaker. The heating substrate was used to simulate 12 μm and 13 μm printing sample temperature changes under ultrasonic amplitudes; print sample tensile strength is higher than 10 μm and 11 μm ultrasonic amplitudes to print the tensile strength of the samples. The ultrasonic amplitude is very big, inside the middle layers of the wire-bonding between areas and a group of a similar simulation temperature, better bonding area, several layers above the ultrasonic vibration are stronger, and the kinetic ability of the molecular chain is greatly enhanced, so the tensile strength of the group with ultrasonic vibration will be stronger at this time.

#### 3.3.2. Influence of Printing Speed and Substrate Temperature on the Mechanical Properties of Printed Samples

Figure 15a,b show the changes in the surface temperature, tensile strength and bending strength, and modulus of the printed samples at different printing speeds under the simulated ultrasonic amplitude of 12 μm with a heated substrate. As can be seen in Figure 15a, with an increase in printing speed the tensile strength of the printed samples of the group with and without a simulated temperature gradually decreased. Compared with the group without a simulated temperature, the tensile strength of the printed samples with a simulated temperature increased by 6.4% at least and by 11.9% at the same speed. In Figure 15b, it can be seen that with an increase in the printing speed no simulation of the temperature of a set of print samples between the bending strength and modulus is reduced, and a set of simulated temperature bending strengths increased after initially decreasing; the bending modulus has no obvious change rule, plus a set of simulated temperature bending strength and modulus than that under the same printing speed without a set of simulated temperature was low, at least 1.9 percent and 0.9 percent, and at most 5.1 percent and 4.5 percent, respectively.

Figure 16 shows the surface temperature changes in the tensile sample when the substrate is heated to simulate the ultrasonic amplitude of 12 μm and the printing speeds of 40 mm/s and 60 mm/s, the section morphology of the printed tensile sample after the mechanical test, and the section morphology of the printed tensile sample after the mechanical test without the heat treatment at the simulated temperature. Through Figure 16a,c it can be seen that the printing of samples that were not heat-treated were in the process of internal wire-bonding between an increase with the decrease in the area as the speed of printing, because the printing speed is smaller, the nozzle is in the current position, the longer the residence time for wire transfer, the longer the silk material in combination with the interface between them, and the higher the temperature. By increasing the bonding interface area and softening time, the polymer chains at the bonding interface can be well-diffused and wound. Through Figure 16b,d, the internal wire samples from the fourth floor to the ninth floor between the combination of the interface bonding area ratio under the same speed without the simulation of high-temperature heat treatment of samples can be observed. After nine samples the adhesive area is lower than without the stimulation of the temperature of heat treatment, because print layers between the simulation higher temperature lead to the bigger wire-bonding process of liquidity; after the ninth layer the distance between the upper layer and the current layer increases, so that the bonding effect between the deposited silk is worse than that without the simulated temperature.

The results of SEM, the mechanical properties, and the simulated temperature curves show that the bending strength and modulus of samples treated with a simulated temperature are lower than those treated without a simulated temperature at the same speed. This is because in the bending test, seen from the middle of the sample, the closer to the pressure head position, the greater the compressive stress, and the farther away from the pressure head position the greater the tensile stress. When the simulated temperature is added, although the middle layers are better bonded, the compressive stress and tensile stress are improved, but the tensile stress is small near the center itself, and the improvement effect is limited. The upper layer bears large compressive stress because the bonding effect is poor, so the compressive stress is greatly reduced, resulting in the reduction in bending strength. At the same speed, the tensile strength increases with the simulated temperature, because in the tensile process the entire tensile interface is subjected to tensile stress, and the bonding effect of the intermediate layer is good, which can greatly increase the tensile capacity; therefore, the tensile strength of the sample is improved.

The bending strength and modulus of the samples at different printing speeds with different ultrasonic amplitudes of 12 μm were higher than those at the same speed and simulated temperature. When the ultrasonic amplitude of 12 μm was added, the surface temperature of the deposited wire was only increased, and the temperature of the whole sample was not increased. Therefore, the wire did not sink significantly in the process of deposition to the middle part. The bonding effect of the sample was almost the same as that of the sample without ultrasonic strengthening when it was hit to the top. At the same time, the ultrasonic vibration made the motion ability of the molecular chain stronger, making the diffusion at the bonding interface faster at the same temperature. Therefore, the bending strength did not decrease but increased compared with the sample without ultrasonic strengthening. Under the same 3D-printing parameters, the tensile strength of the sample printed with a 12 μm ultrasonic amplitude is higher than that of the sample printed with a simulated temperature, because the ultrasonic vibration not only increases the temperature at the bonding interface but also significantly increases the motion ability of the internal molecular chain, so that the diffusion and entanglement between the internal molecular chains are much faster than the simple temperature effect. The diffusion and entanglement of the molecular chain increase the force capacity of the sample per unit area, so the tensile strength is higher.

#### 3.3.3. The Effects of Layer Thickness and Substrate Temperature on Mechanical Properties

Figure 17a,b show the variation curves of the tensile strength as well as bending strength and modulus of printed mechanical samples under different printing layer thicknesses under the simulated ultrasonic amplitude of 12 μm on a heated substrate. As can be seen from Figure 17a, the tensile strength of the printed samples with and without a simulated temperature decreases gradually with the increase in printing layer thickness. The tensile strength of the sample printed with a simulated temperature increased at least 5.7 % and at most 9.8 % compared with the sample printed with the same layer thickness without a simulated temperature. It can be seen from Figure 17b that with an increase in printing layer thickness, the bending strength and modulus of the printed samples with and without a simulated temperature decrease gradually. The bending strength and modulus of the printed samples with a simulated temperature decrease by more than 3.5% and 3.5% at least, and decrease by 4.5% and 5.7% at most compared with those without a simulated temperature at the same layer thickness. As the printing layer thickness increases, the tensile strength of samples without a simulated temperature gradually decreases.

## 4. Conclusions

In order to improve the mechanical properties of FDM parts, this paper put forward the method of ultrasonic-strengthening 3D-printing parts in the process of forming. An ultrasonic-vibration-assisted enhanced FDM 3D printer was developed. The effects of process parameters such as ultrasonic amplitude, printing layer thickness, and printing speed on the mechanical properties of the sample were studied. The mechanism of process parameters on the mechanical properties of the sample was explored through microscopic characterization and thermal stability analysis. With the increase in ultrasonic amplitude, the tensile strength and bending modulus of the sample increased gradually, the bending strength increased first and then decreased, and the bonding quality between the internal rasters became better and better. For various printing speeds and printing layer thicknesses, the mechanical properties of the sample with an ultrasonic amplitude were significantly improved, and the bonding effect between the internal rasters was better. Under the same 3D-printing parameters, when printing to the same number of layers the surface temperature of the samples with ultrasonic vibration was higher than that of the samples without ultrasonic vibration. The experimental results showed that the mechanical properties of printed samples were improved by ultrasonic vibration, and that the temperature increased in printed samples. Ultrasonic vibration substrate-assisted FDM improved the mechanical properties of forming samples significantly, which can broaden the application and development of low-cost FDM 3D-printing technology in the field of engineering.

## Figures and Tables

**Figure 1 polymers-14-00904-f001:**
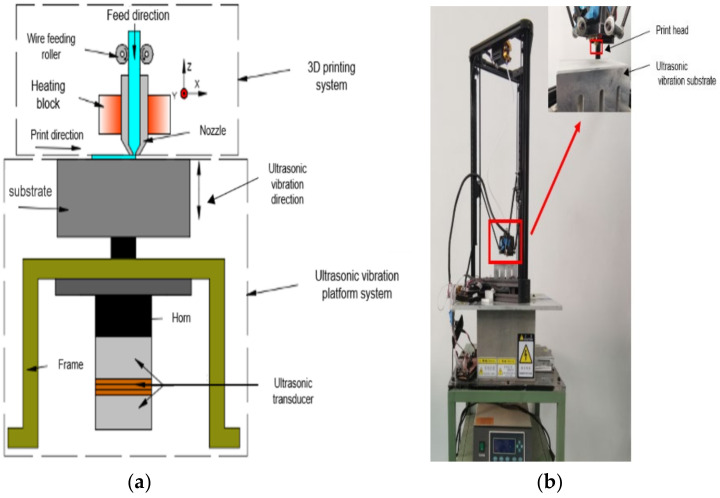
Schematic diagram (**a**) and real printer (**b**) of ultrasonic-assisted FDM.

**Figure 2 polymers-14-00904-f002:**
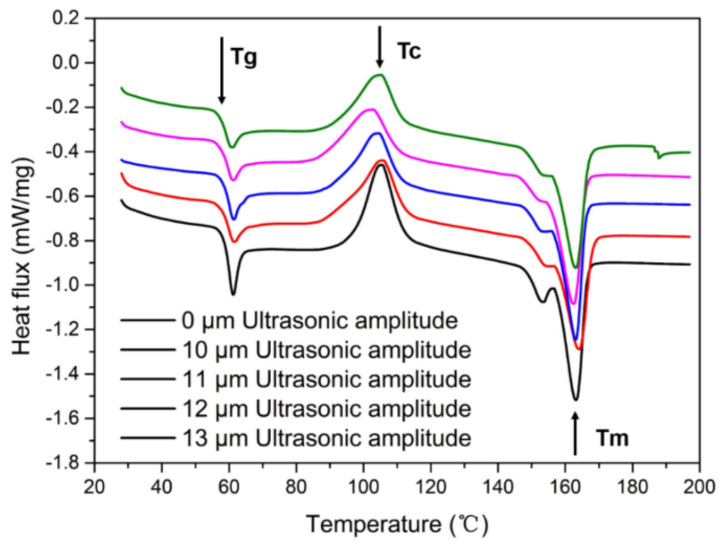
DSC curves of samples under different ultrasonic amplitudes.

**Figure 3 polymers-14-00904-f003:**
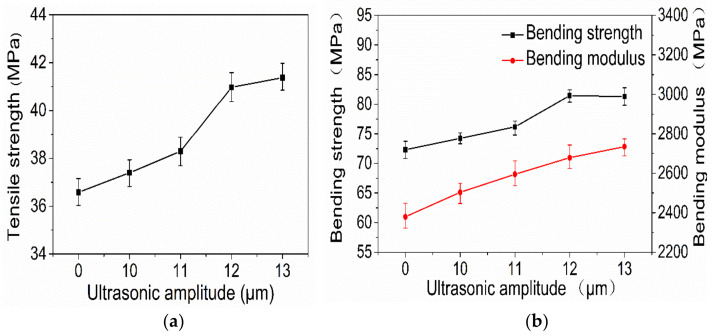
Tensile strength (**a**) and bending properties (**b**) of samples at various amplitudes.

**Figure 4 polymers-14-00904-f004:**
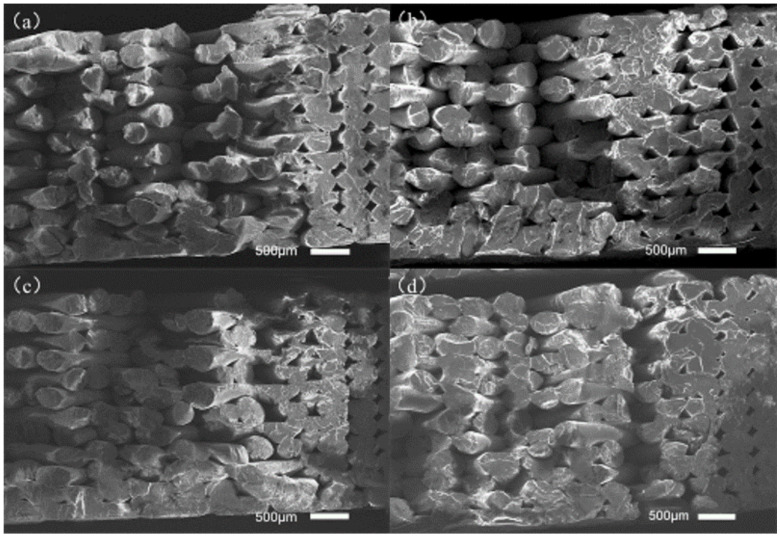
Section morphology of tensile samples under different ultrasonic amplitudes: (**a**) 0 μm, (**b**) 10 μm, (**c**) 11 μm, and (**d**) 12 μm ultrasound-enhanced samples, respectively.

**Figure 5 polymers-14-00904-f005:**
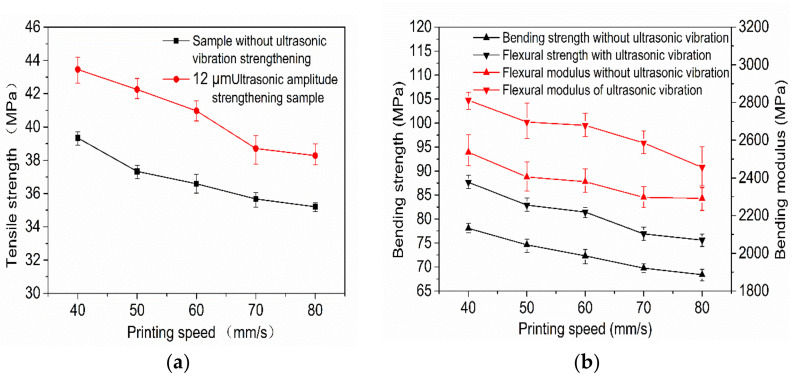
Tensile strength (**a**) and bending properties (**b**) of samples at various speeds.

**Figure 6 polymers-14-00904-f006:**
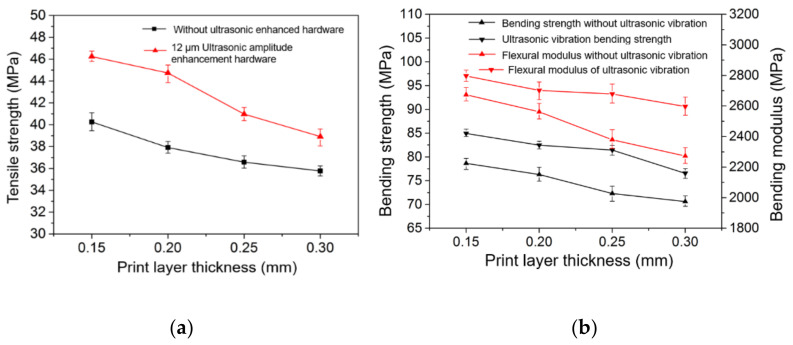
Tensile strength (**a**) and bending properties (**b**) of samples at various layers.

**Figure 7 polymers-14-00904-f007:**
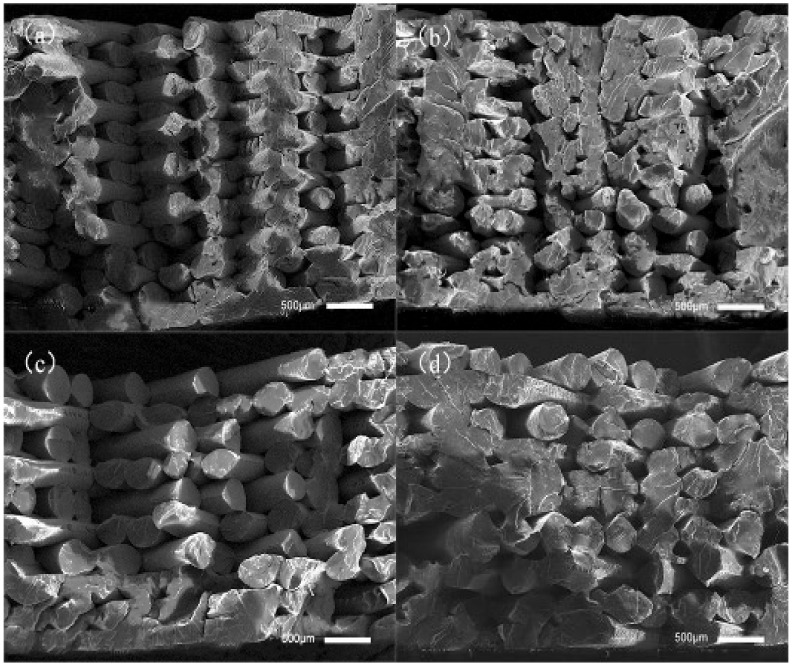
Section morphologies of tensile samples at different layer thicknesses. (**a**,**c**) were 0.20 mm, 0.30 mm layer thickness without ultrasonic vibration sample; (**b**,**d**) were 0.20 mm, 0.30 mm layer thickness with an ultrasonic amplitude sample of 12 μm.

**Figure 8 polymers-14-00904-f008:**
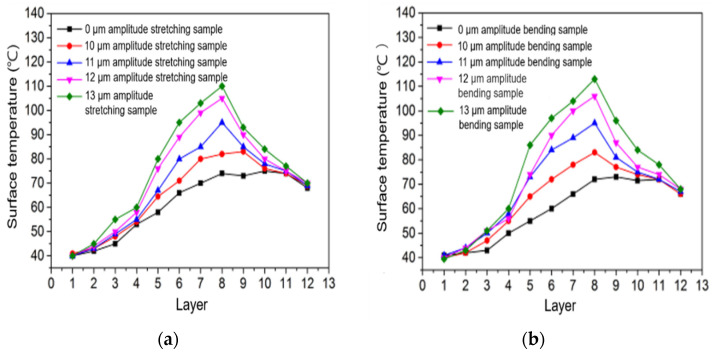
Temperature changes in tensile samples (**a**) and bending samples (**b**) at various amplitudes.

**Figure 9 polymers-14-00904-f009:**
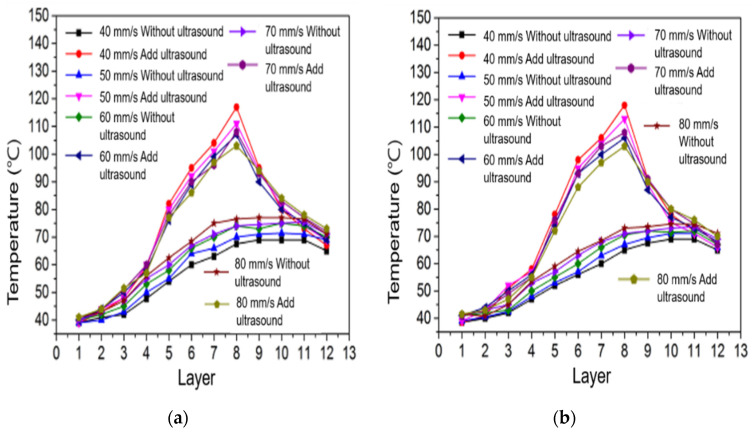
Temperature changes in tensile samples (**a**) and bending samples (**b**) at various speeds.

**Figure 10 polymers-14-00904-f010:**
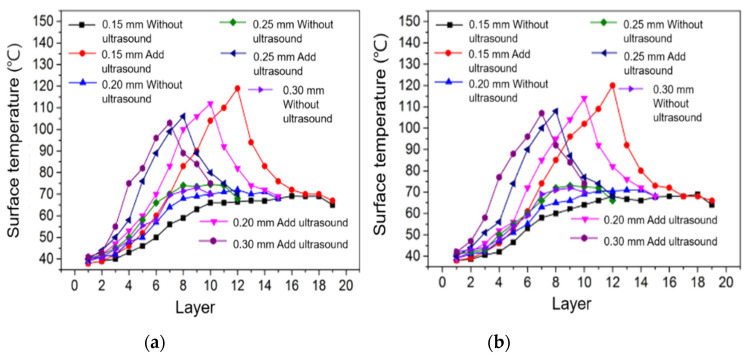
Temperature changes in tensile samples (**a**) and bending samples (**b**) at various layer thicknesses.

**Figure 11 polymers-14-00904-f011:**
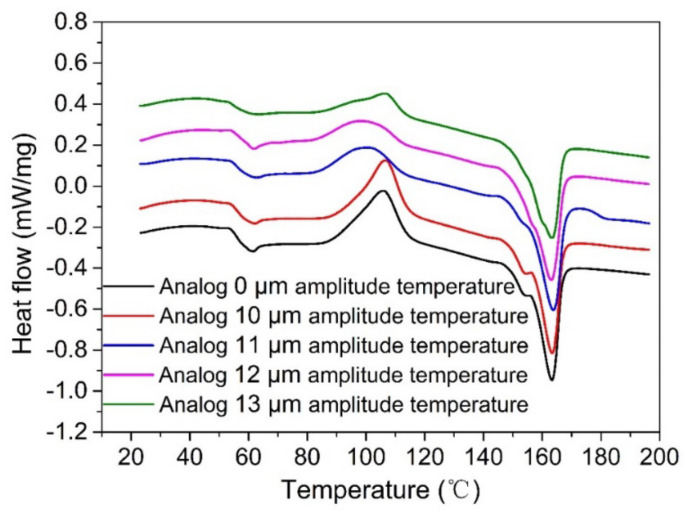
DSC curve simulating temperature changes in samples at different amplitudes.

**Figure 12 polymers-14-00904-f012:**
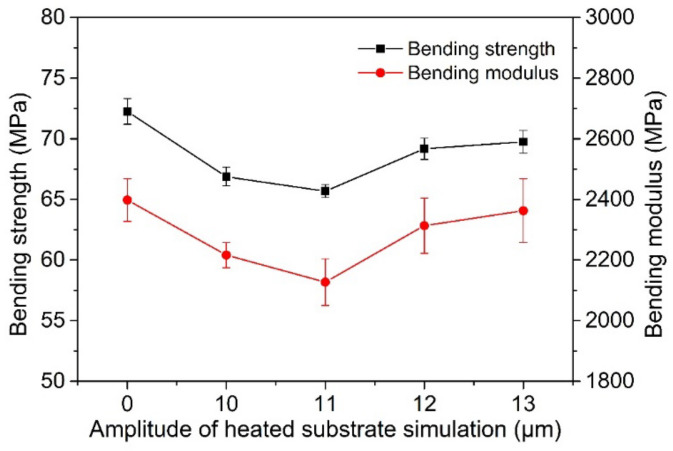
The bending strength and modulus of printed samples at different amplitudes and temperatures.

**Figure 13 polymers-14-00904-f013:**
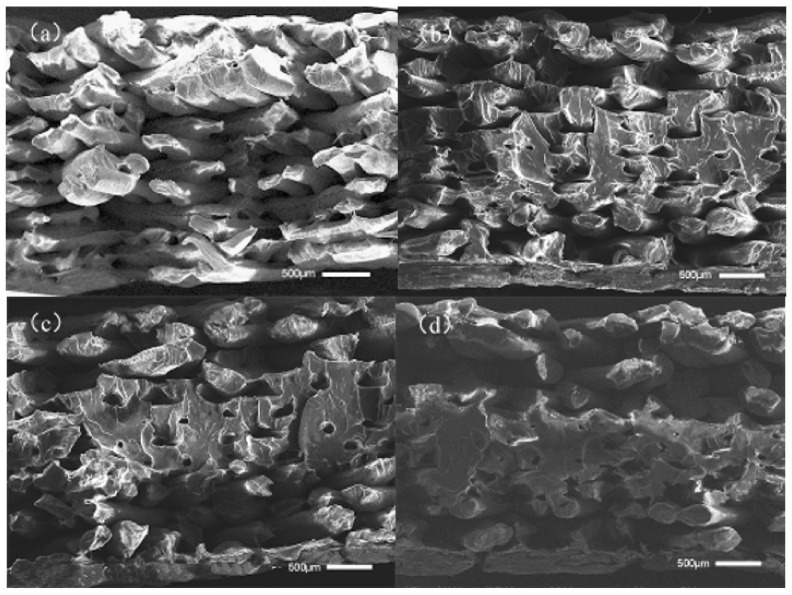
Simulated section morphology of printed samples at different amplitudes and temperatures. (**a**–**d**) are the printed samples of the heated substrate simulated at an ultrasonic amplitude temperature of 0 μm, 10 μm, 11 μm, and 12 μm, respectively.

**Figure 14 polymers-14-00904-f014:**
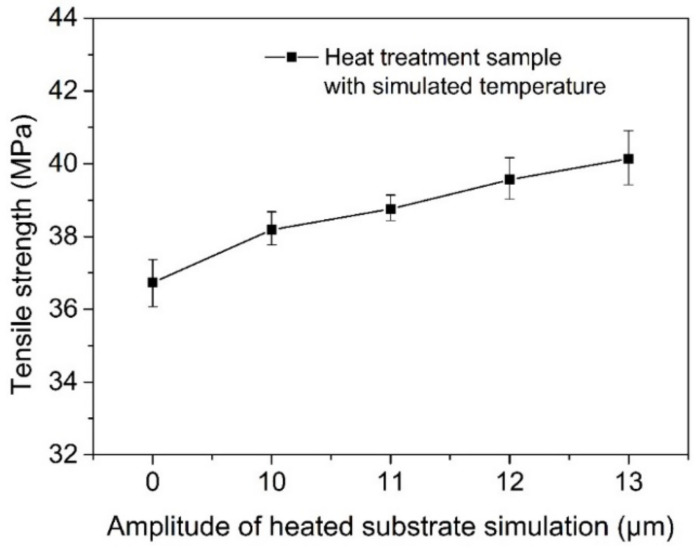
The tensile strength of printed samples at different amplitudes and temperatures.

**Figure 15 polymers-14-00904-f015:**
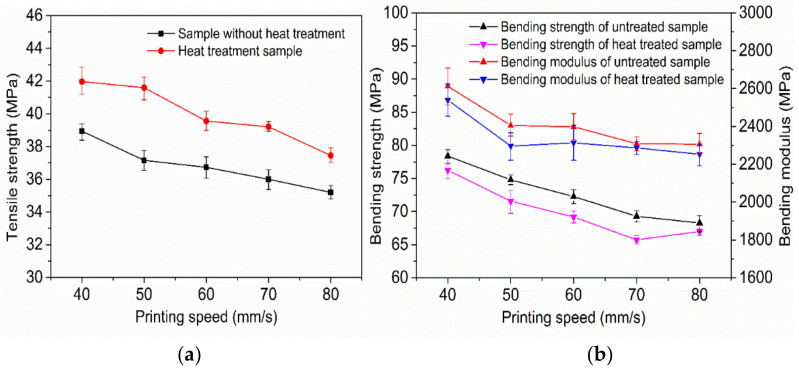
The tensile strength (**a**) and bending properties (**b**) of samples at various simulated speeds and temperatures.

**Figure 16 polymers-14-00904-f016:**
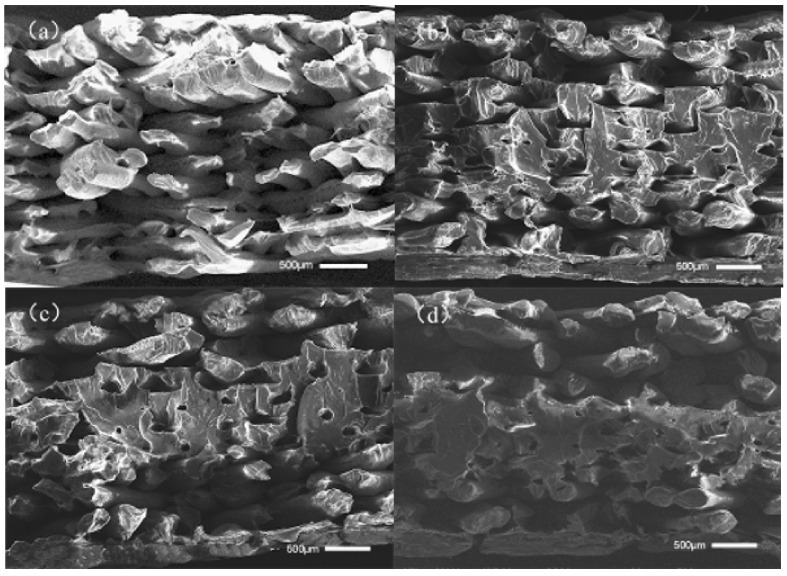
Simulated section morphology of printed samples at different speeds and temperatures. (**a**,**c**) are the fracture sections of the tensile sample at the printing speeds of 40 mm/s and 60 mm/s; (**b**,**d**) are the fracture sections of the tensile sample at the speeds of 40 mm/s and 60 mm/s with the addition of a simulated temperature.

**Figure 17 polymers-14-00904-f017:**
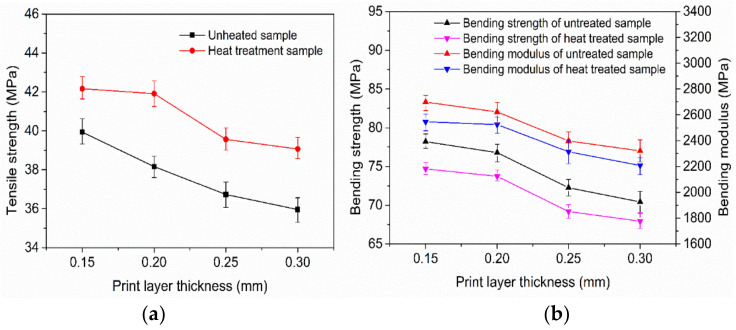
The tensile strength (**a**) and bending properties (**b**) of samples at various simulated layer thicknesses and temperatures.

**Table 1 polymers-14-00904-t001:** Experiment on the influence of processing parameters on the mechanical properties.

Test Group	Serial Number	Ultrasonic Amplitude A (μm)	Print Speed v (mm/s)	Printing Layer Thickness LT (mm)
A	1	0	60	0.25
	2	10	60	0.25
	3	11	60	0.25
	4	12	60	0.25
	5	13	60	0.25
B	6	12	40	0.25
	7	12	50	0.25
	8	12	60	0.25
	9	12	70	0.25
	10	12	80	0.25
	11	0	40	0.25
	12	0	50	0.25
	13	0	60	0.25
	14	0	70	0.25
	15	0	80	0.25
C	16	12	60	0.15
	17	12	60	0.20
	18	12	60	0.25
	19	12	60	0.30
	20	0	60	0.15
	21	0	60	0.20
	22	0	60	0.25
	23	0	60	0.30

**Table 2 polymers-14-00904-t002:** Experiment on the thermal influence of ultrasonic strengthening samples.

Test Group	Serial Number	Ultrasonic Amplitude A (μm)	Print Speed v (mm/s)	Printing Layer Thickness LT (mm)
A	1	0	60	0.25
	2	10	60	0.25
	3	11	60	0.25
	4	12	60	0.25
	5	13	60	0.25
B	6	12	40	0.25
	7	12	50	0.25
	8	12	60	0.25
	9	12	70	0.25
	10	12	80	0.25
C	11	12	60	0.15
	12	12	60	0.20
	13	12	60	0.25
	14	12	60	0.30

**Table 3 polymers-14-00904-t003:** Simulate amplitude experiments on the effects of temperature on the mechanical properties.

Test Group	Serial Number	Simulate Amplitude A (μm)	Print Speed v (mm/s)	Printing Layer Thickness LT (mm)
A	1	0	60	0.25
	2	10	60	0.25
	3	11	60	0.25
	4	12	60	0.25
	5	13	60	0.25
B	6	12	40	0.25
	7	12	50	0.25
	8	12	60	0.25
	9	12	70	0.25
	10	12	80	0.25
C	11	12	60	0.15
	12	12	60	0.20
	13	12	60	0.25
	14	12	60	0.30

## Data Availability

The data presented in this study are available on request from the corresponding author.

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
