# Peer review of "Influence Mechanism of Ultrasonic Vibration Substrate on Strengthening the Mechanical Properties of Fused Deposition Modeling"

_polymers, 2022, doi:10.3390/polym14050904_

Round 1

Reviewer 1 Report

Manuscript ID: polymers-1598881 Title: Ultrasonic assisted fused deposition modeling. The reviewer queries is as follows:

  1. The title is very general and not clear.  
  2.  Include some important numerical results in the abstract.
  3.  In my opinion, Fig. 1 is not necessary and this can be explained through text.
  4.  The equations are to be properly cited.
  5.  What the authors are trying to explain from the SEM images presented in Fig. 7? And the difference between SEM images presented in Fig. 7 and Fig. 13 and Fig. 16 need to be explained.
  6. 6. The conclusion must be precise and to the point.        

Author Response

Dear Editors and Reviewers:

Thanks very much for the reviewer’s comments concerning our manuscript. Those comments are all valuable and very helpful for revising. We have studied the comments carefully and made corrections which we hope meet with approval. Revised portion are marked in the paper. The main corrections in the paper and the response to the reviewer’s comments are as following:

Point 1: The title is very general and not clear.

Response 1: Thanks for the reviewer’s suggestion. We have revised the title as following.

Influence mechanism of ultrasonic vibration substrate on strengthening mechanical properties of fused deposition modelling

Point 2: Include some important numerical results in the abstract.

Response 2: Thanks for the reviewer’s suggestion. We have revised the abstract section as following.

Fused deposition modeling is the most widely used 3D printing technology with the advantage of accessible forming process. However, the poor mechanical properties of the forming parts limit its application in engineering. Here, a new ultrasonic assisted fused deposition modeling 3D printing method is proposed to improve the mechanical properties of formed parts. The effects of ultrasonic vibration substrate process parameters and printing process parameters on the tensile and bending properties of formed samples were studied. The tensile strength and bending strength of the samples printed with 12 μm ultrasonic amplitude can be increased by 13.2% and 12.6%, respectively, compared with those printed without ultrasonic vibration. The influence mechanism of ultrasonic vibration on mechanical properties was studied by microscopic characterization and in-situ infrared monitoring experiments. During the printing process, the ultrasonic vibration and temperature increasing employed via the ultrasonic substrate can reduce the pore defects inside the sample. Mechanical properties of FDM forming samples can be controlled by adjusting ultrasonic assisted process parameters, which can broaden the application of 3D printing.

Point 3: In my opinion, Fig. 1 is not necessary and this can be explained through text.

Response 3: The authors sincerely thanks for the reviewer’s suggestion. We think Fig. 1 may be more visual to show the working principle of our home-made printer. We sincerely hope you could allow us to keep Fig.1 in the text. And with the suggestion of other reviewer, we also added a picture of the printer to Fig. 1 to illustrate it. We have revised the Fig. 1.

Point 4: The equations are to be properly cited.

Response 4: Thanks for the reviewer’s suggestion. We have revised the equations by citing related reference in the manuscript.

Point 5: What the authors are trying to explain from the SEM images presented in Fig. 7? And the difference between SEM images presented in Fig. 7 and Fig. 13 and Fig. 16 need to be explained.

Response 5: The authors sincerely thanks for the reviewer’s suggestion.

Figure 7 shows section morphologies of tensile samples at different layer thicknesses. (a) and (c) was 0.20 mm, 0.30 mm layer thickness without ultrasonic vibration sample, (b) and (d) were 0.20 mm, 0.30 mm layer thickness with ultrasonic amplitude sample of 12 μm. The cross section of the sample is denser after ultrasonic application. We tried to express the effects of layer thickness and ultrasonic vibrations during the FDM processing with Figure 7.

Figure 13 shows simulated section morphology of printed samples at different amplitude and temperature. (a), (b), (c), and (d) are the printed samples of the heated substrate simulated at ultrasonic amplitude temperature of 0 μm, 10 μm, 11 μm, and 12 μm, respectively. We tried to express that the bonding quality of internal rasters of the sample with simulated ultrasonic amplitude temperature was better, and there were almost no gaps between some rasters. 

Figure 16 shows simulated section morphology of printed samples at different speeds and temperatures. (a) and (c) are the fracture sections of the tensile sample at the printing speed of 40 mm/s and 60 mm/s; (b) and (d) are the fracture sections of the tensile sample at the speed of 40 mm/s and 60 mm/s with addition of simulated temperature. The cross section of the sample is denser with ultrasonic application and slower printing speed. We tried to express the effects of printing speed and simulated temperature during the FDM processing with Figure 16.

Point 6: The conclusion must be precise and to the point.

Response 6: Thanks for the reviewer’s suggestion. We have revised the conclusion as following.

In order to improve the mechanical properties of FDM parts, this paper put forward the method of ultrasonic strengthening 3D printing parts in the process of forming. Ultra-sonic vibration-assisted enhanced FDM 3D printer was developed. The effects of process parameters such as ultrasonic amplitude, printing layer thickness and printing speed on the mechanical properties of the sample were studied. The mechanism of process parameters on the mechanical properties of the sample was explored through microscopic characterization and thermal stability analysis. With the increase of ultrasonic amplitude, the tensile strength and bending modulus of the sample increased gradually, the bending strength increased first and then decreased, and the bonding quality between the internal rasters got better and better. For various printing speeds and printing layer thickness, the mechanical properties of the sample with ultrasonic amplitude were significantly improved, and the bonding effect between the internal rasters was better. Under the same 3D printing parameters, when printing to the same number of layers, the surface temperature of the samples with ultrasonic vibration was higher than that of the samples without ultrasonic vibration. The experimental results showed that the mechanical properties of printed samples were improved by ultrasonic vibration and the temperature increased of printed samples. Ultrasonic vibration substrate assisted FDM improved the mechanical properties of forming samples significantly, which can broaden the application and development of low-cost FDM 3D printing technology in the field of engineering.

Reviewer 2 Report

  1. The article needs to English grammar editing. Formal writing is almost always written in the third person or passive tenses. There are too many cases in the manuscript, where the authors have used "We".
  2. The abbreviation should be avoided in abstract, or the full writing should be given when it appears first time.
  3. The introduction is suggested to be enriched, such as the research progress in ultrasonic process.
  4. Line 88, Page 3: “isdesigned” should be separated. The same problem for “â–³??is”(Line 201). Please proofread the whole manuscript carefully.
  5. The authors should use same symbol to denote the printing speed in Table 1 and Table 2.
  6. It would be much clearer and the readability can be improved if the content “Lines 155-174” was presented in a table.
  7. The unit of heat flux in figure 2 is W/g, however, the SI of heat flux is W/m2 or J/s.
  8. In figures 3, 5, 6, 12, 14, 15, 17, the unit should be MPa, but not Mpa. The authors should proofread the whole manuscript carefully to avoid such mistakes or typos.
  9. Some figures are not clear, such as figures 5, 10. The quality of these figures should be improved and original picture (high-resolution) should be used.
  10. In figure 15, the subfigure (g) should be (a).
  11. It is suggested to simplify the conclusion, the current one is too long. Please only preset the most important findings, but not repeat the results.
  12. In the Reference, the reference number was repeated.

Author Response

Dear Editors and Reviewers:

Thanks very much for the reviewer’s comments concerning our manuscript. Those comments are all valuable and very helpful for revising. We have studied the comments carefully and made corrections which we hope meet with approval. Revised portions are marked in the paper. The main corrections in the paper and the response to the reviewer’s comments are as following:

 Point 1: The article needs to English grammar editing. Formal writing is almost always written in the third person or passive tenses. There are too many cases in the manuscript, where the authors have used "We".

Response 1: The authors sincerely thanks for the reviewer’s suggestion. We have rechecked our grammar mistakes and corrected the usage of person tense.

Point 2: The abbreviation should be avoided in abstract, or the full writing should be given when it appears first time.

Response 2: Thanks for the reviewer’s suggestion. We checked the usage of abbreviations in the text and made the relative corrections.

Herein, a new ultrasonic assisted fused deposition modeling 3D printing method was proposed to improve the mechanical properties of formed parts.

Point 3: The introduction is suggested to be enriched, such as the research progress in ultrasonic process.

Response 3: Thanks for the reviewer’s suggestion. We have added the research progress in ultrasonic process as following.

Gunduz et al. conducted relevant research on ultrasonic vibration assisted 3D print-ing of high viscosity materials, introduced high amplitude ultrasonic vibration into the nozzle, and finally commercial polymer clay with viscosity up to 14000 pa · s[23]. Richard et al. found that ultrasonic treatment can effectively improve the surface finish, reduce the porosity under the surface, increase the surface hardness and improve the fatigue perfor-mance of 3D printed metal parts[24, 25]. Alireza et al. explored the effect of ultrasonic vi-bration print head on the bonding interface strength of printing wire produced by FDM[26]. When ultrasonic vibration was added in the process of printing wire, the inter-layer bonding strength increased by up to 10%.

Point 4: Line 88, Page 3: “isdesigned” should be separated. The same problem for “â–³??is”(Line 201). Please proofread the whole manuscript carefully.

Response 4: Thanks for the reviewer’s suggestion. We proofread the whole manuscript and corrected the format errors.

Point 5: The authors should use same symbol to denote the printing speed in Table 1 and Table 2.

Response 5: Thanks for the reviewer’s suggestion. We have used the same symbols to represent the printing speeds in Tables 1 and 2 as following.

Table 1 Experimental factors and levels of ultrasonic assisted strengthening parameters

Test group

Serial number

Ultrasonic amplitude

A(μm)

Print speed

v(mm/s)

printing

layer thickness

LT(mm)

Table 2 Experimental factors and levels of ultrasonic assisted strengthening parameters

Test group

Serial number

Ultrasonic amplitude

A(μm)

Print speed

v(mm/s)

printing layer thickness

LT(mm)

Point 6: It would be much clearer and the readability can be improved if the content “Lines 155-174” was presented in a table.

Response 6: Thanks for the reviewer’s suggestion. We have converted the contents “Lines 155-174” into tables with text descriptions as following.

Group A used heated substrate to simulate the change of surface temperature during the printing process of tensile samples and bending samples under different ultrasonic amplitudes, and printed tensile samples and bending samples. The actual ultrasonic amplitude is 0 μ m。Group B used heated substrate to simulate the temperature change of sample surface during printing under fixed ultrasonic amplitude and different printing speed, and printed tensile sample and bending sample; at the same time, experimental group B was added to the control group without simulated temperature heat treatment, other experimental parameters were the same as those of experimental group B. The actual ultrasonic amplitude of the two groups was 0 μm。In experimental group C, the surface temperature changes of samples during printing, tensile samples and bending samples were simulated with heated substrate under fixed ultrasonic amplitude and different printing layer thickness; at the same time, group C was added to the control test group without simulated temperature heat treatment, other experimental parameters were the same as those of experimental group C ,and the actual ultrasonic amplitude of the two groups was 0 μ m。Table 3 shows the arrangement of printing test parameters for each group.Table 3 Experimental factors and levels of ultrasonic assisted strengthening parameters

Test group

Serial number

Ultrasonic amplitude

A(μm)

Print speed

v(mm/s)

printing layer thickness

LT(mm)

A

1

0

60

0.25

2

10

60

0.25

3

11

60

0.25

4

12

60

0.25

5

13

60

0.25

B

6

12

40

0.25

7

12

50

0.25

8

12

60

0.25

9

12

70

0.25

10

12

80

0.25

C

11

12

60

0.15

12

12

60

0.20

13

12

60

0.25

14

12

60

0.30

Point 7: The unit of heat flux in figure 2 is W/g, however, the SI of heat flux is W/m2 or J/s.

Response 7: Thanks for the reviewer’s suggestion. We have modified the heat flow units.

Point 8: In figures 3, 5, 6, 12, 14, 15, 17, the unit should be MPa, but not Mpa. The authors should proofread the whole manuscript carefully to avoid such mistakes or typos.

 Response 8: Thanks for the reviewer’s suggestion. We have revised the writing of all megapascals.

Point 9: Some figures are not clear, such as figures 5, 10. The quality of these figures should be improved and original picture (high-resolution) should be used.

Response 9: Thanks for the reviewer’s suggestion. We have optimized the images and used higher resolution images.

Figure 10. Temperature changes of tensile samples (a) and bending samples (b) at various layer thickness.

Point 10: In figure 15, the subfigure (g) should be (a).

Response 10: Thanks for the reviewer’s suggestion. We have corrected the mistakes.

Point 11: It is suggested to simplify the conclusion, the current one is too long. Please only preset the most important findings, but not repeat the results.

Response 11: Thanks for the reviewer’s suggestion. We have revised the Conclusion section as following.

In order to improve the mechanical properties of FDM parts, this paper put forward the method of ultrasonic strengthening 3D printing parts in the process of forming. Ultra-sonic vibration-assisted enhanced FDM 3D printer was developed. The effects of process parameters such as ultrasonic amplitude, printing layer thickness and printing speed on the mechanical properties of the sample were studied. The mechanism of process parameters on the mechanical properties of the sample was explored through microscopic characterization and thermal stability analysis. With the increase of ultrasonic amplitude, the tensile strength and bending modulus of the sample increased gradually, the bending strength increased first and then decreased, and the bonding quality between the internal rasters got better and better. For various printing speeds and printing layer thickness, the mechanical properties of the sample with ultrasonic amplitude were significantly improved, and the bonding effect between the internal rasters was better. Under the same 3D printing parameters, when printing to the same number of layers, the surface temperature of the samples with ultrasonic vibration was higher than that of the samples without ultrasonic vibration. The experimental results showed that the mechanical properties of printed samples were improved by ultrasonic vibration and the temperature increased of printed samples. Ultrasonic vibration substrate assisted FDM improved the mechanical properties of forming samples significantly, which can broaden the application and development of low-cost FDM 3D printing technology in the field of engineering.

Point 12: In the Reference, the reference number was repeated.

Response 12: Thanks for the reviewer’s suggestion. We have revised the reference number.

Reviewer 3 Report

The work presented for evaluation, entitled Ultrasonic assisted fused deposition modeling is typically engineering rather than scientific in nature. It concerns the influence of a number of parameters of ultrasonically assisted 3D printing on the mechanical properties of polymers from the group of polyesters. It has been shown that by treating the freshly printed product with ultrasound a slight improvement in properties is achieved in this case. It should be added that the authors of the previous work on 3D printing with ultrasound for a filament from the styrene group achieved a clear increase in the mechanical properties of the product. For this reason, the work can be classified as a very large number of papers on the interaction of ultrasounds with polymers.

However, the authors did not present too many areas of application of ultrasound to plimers and they limited themselves to a small number of examples concerning the 3D printing itself.

They did not take up the issue of the impact of ultrasound on polymers, including the differences in the absorption properties of different polymers (plus additive), especially since their research concerns not pure polymers, but commercial PLA filament.

It should be added that, despite the large amount of literature, the research presented is also important for application reasons.

When reading the manuscript, a number of editorial shortcomings were noticed, e.g. in lines 333, 486 and others.

Author Response

Dear Editors and Reviewers:

 Thanks very much for the reviewer’s comments concerning our manuscript. Those comments are all valuable and very helpful for revising. We have studied the comments carefully and made corrections which we hope meet with approval. Revised portions are marked in the paper. The main corrections in the paper and the response to the reviewer’s comments are as following:

 Point 1: The work presented for evaluation, entitled Ultrasonic assisted fused deposition modeling is typically engineering rather than scientific in nature. It concerns the influence of a number of parameters of ultrasonically assisted 3D printing on the mechanical properties of polymers from the group of polyesters. It has been shown that by treating the freshly printed product with ultrasound a slight improvement in properties is achieved in this case. It should be added that the authors of the previous work on 3D printing with ultrasound for a filament from the styrene group achieved a clear increase in the mechanical properties of the product. For this reason, the work can be classified as a very large number of papers on the interaction of ultrasounds with polymers.

However, the authors did not present too many areas of application of ultrasound to plimers and they limited themselves to a small number of examples concerning the 3D printing itself.

They did not take up the issue of the impact of ultrasound on polymers, including the differences in the absorption properties of different polymers (plus additive), especially since their research concerns not pure polymers, but commercial PLA filament.

It should be added that, despite the large amount of literature, the research presented is also important for application reasons.

Response 1: Thanks for the reviewer’s suggestion. We have supplemented the introduction according to the content you mentioned about the influence of ultrasound on processing as following.

Gunduz et al. conducted relevant research on ultrasonic vibration assisted 3D print-ing of high viscosity materials, introduced high amplitude ultrasonic vibration into the nozzle, and finally commercial polymer clay with viscosity up to 14000 pa·s[23]. Richard et al. found that ultrasonic treatment can effectively improve the surface finish, reduce the porosity under the surface, increase the surface hardness and improve the fatigue perfor-mance of 3D printed metal parts[24, 25]. Alireza et al. explored the effect of ultrasonic vi-bration print head on the bonding interface strength of printing wire produced by FDM[26]. When ultrasonic vibration was added in the process of printing wire, the inter-layer bonding strength increased by up to 10%.

We also revised the conclusion part as following:

In order to improve the mechanical properties of FDM parts, this paper put forward the method of ultrasonic strengthening 3D printing parts in the process of forming. Ultra-sonic vibration-assisted enhanced FDM 3D printer was developed. The effects of process parameters such as ultrasonic amplitude, printing layer thickness and printing speed on the mechanical properties of the sample were studied. The mechanism of process param-eters on the mechanical properties of the sample was explored through microscopic char-acterization and thermal stability analysis. With the increase of ultrasonic amplitude, the tensile strength and bending modulus of the sample increased gradually, the bending strength increased first and then decreased, and the bonding quality between the internal rasters got better and better. For various printing speeds and printing layer thickness, the mechanical properties of the sample with ultrasonic amplitude were significantly im-proved, and the bonding effect between the internal rasters was better. Under the same 3D printing parameters, when printing to the same number of layers, the surface temperature of the samples with ultrasonic vibration was higher than that of the samples without ul-trasonic vibration. The experimental results showed that the mechanical properties of printed samples were improved by ultrasonic vibration and the temperature increased of printed samples. Ultrasonic vibration substrate assisted FDM improved the mechanical properties of forming samples significantly, which can broaden the application and de-velopment of low-cost FDM 3D printing technology in the field of engineering.

Point 2:When reading the manuscript, a number of editorial shortcomings were noticed, e.g. in lines 333, 486 and others.

Response 2: Thanks for the reviewer’s suggestion. We have proofread the manuscript and corrected the editorial mistakes as following.

The test parts were printed by a homemade ultrasonic vibration-assisted FDM ad-ditive manufacturing device. The 3D printing material is PLA (Hangzhou Xianlin 3D Science and Technology Co., Ltd, China). The tensile test sample is designed by non-standard design. The shape of the sample is designed as a strip, and the size of the sample is 70 mm×8 mm×3 mm. The shape of the bent sample is designed by ISO 178:2001, and the size of the sample is 60 mm×10 mm×3 mm.

Where Xc is the crystallinity, â–³Hm is the enthalpy of melting, â–³Hc is the enthalpy of cold crystallization, â–³Htheory is the theoretical melting enthalpy of 100% crystallization of the tested sample, and the theoretical melting enthalpy of PLA is 93.7 J/g [28].

Reviewer 4 Report

In this study, the authors have performed a research to study the mechanical properties of 3D-printed PLA parts. The idea and the obtained results are interesting and I propose this paper for publication after addressing the following issues and suggestions:

  1. The paper is well-organized and each section has been well defined.

  1. Introduction: This section is well defined, however, the literature review is not sufficient. I propose to add some more references and more explanation through the problematic of the work.

In line 25-38: I propose you to consider the following reference for the application of 3D printing in biomedical:

https://doi.org/10.3390/polym13244442

In line 35-36: I propose the following references for the effect of process parameters on the characteristics of 3D printed parts:

https://doi.org/10.3390/thermo1030021

https://doi.org/10.1108/RPJ-11-2019-0300

  1. In Figure 1: I propose the authors to add a photo of the setup beside the schematic that they have included in the paper.
  2. In Table 1: How you have considered the condition of printing ? Is it based on any previous study or researches by other scholars ?
  3. In page 6 line 203: The value of the theoretical crystallization is correct, however, it is required to mention the associated reference you have used.
  4. Figure 3, 5, 6: The quality of the figure is not good and it is not clear.
  5. I propose to include some explanation of the conclusion through the discussion part. It is suggested to perform a summary and findings of the work.

In general, the paper is well written and it is worthy of publication.

Author Response

Dear Editors and Reviewers:

 Thanks very much for the reviewer’s comments concerning our manuscript. Those comments are all valuable and very helpful for revising. We have studied the comments carefully and made corrections which we hope meet with approval. Revised portions are marked in the paper. The main corrections in the paper and the response to the reviewer’s comments are as following:

 Point 1: The paper is well-organized and each section has been well defined.

Response 1: Thank you for your affirmation. We will continue to work hard to improve the paper.

Point 2: Introduction: This section is well defined, however, the literature review is not sufficient. I propose to add some more references and more explanation through the problematic of the work.

In line 25-38: I propose you to consider the following reference for the application of 3D printing in biomedical:

https://doi.org/10.3390/polym13244442

In line 35-36: I propose the following references for the effect of process parameters on the characteristics of 3D printed parts:

https://doi.org/10.3390/thermo1030021

https://doi.org/10.1108/RPJ-11-2019-0300

Response 2: Thanks for the reviewer’s suggestion. We have added the references.

Point 3: In Figure 1: I propose the authors to add a photo of the setup beside the schematic that they have included in the paper.

Response 3: Thanks for the reviewer’s suggestion. We have modified Figure 1.

Point 4: In Table 1: How you have considered the condition of printing ? Is it based on any previous study or researches by other scholars ?

Response 4: The authors sincerely thanks for the reviewer’s suggestion. We chose considered the condition of printing through some trial and error.

Point 5: In page 6 line 203: The value of the theoretical crystallization is correct, however, it is required to mention the associated reference you have used.

Response 5: The authors sincerely thanks for the reviewer’s suggestion. We have added the reference.

Point 6: Figure 3, 5, 6: The quality of the figure is not good and it is not clear.

Response 6: Thanks for the reviewer’s suggestion. We have modified Figure 3,5,6 .                                                     

Point 7: I propose to include some explanation of the conclusion through the discussion part. It is suggested to perform a summary and findings of the work.

Response 7: Thanks for the reviewer’s suggestion. We have revised the Conclusion section as following.

In order to improve the mechanical properties of FDM parts, this paper put forward the method of ultrasonic strengthening 3D printing parts in the process of forming. Ultra-sonic vibration-assisted enhanced FDM 3D printer was developed. The effects of process parameters such as ultrasonic amplitude, printing layer thickness and printing speed on the mechanical properties of the sample were studied. The mechanism of process parameters on the mechanical properties of the sample was explored through microscopic characterization and thermal stability analysis. With the increase of ultrasonic amplitude, the tensile strength and bending modulus of the sample increased gradually, the bending strength increased first and then decreased, and the bonding quality between the internal rasters got better and better. For various printing speeds and printing layer thickness, the mechanical properties of the sample with ultrasonic amplitude were significantly improved, and the bonding effect between the internal rasters was better. Under the same 3D printing parameters, when printing to the same number of layers, the surface temperature of the samples with ultrasonic vibration was higher than that of the samples without ultrasonic vibration. The experimental results showed that the mechanical properties of printed samples were improved by ultrasonic vibration and the temperature increased of printed samples. Ultrasonic vibration substrate assisted FDM improved the mechanical properties of forming samples significantly, which can broaden the application and development of low-cost FDM 3D printing technology in the field of engineering.

Round 2

Reviewer 1 Report

The revised manuscript is improved and the authors are responded well for the queries raised by me and other reviewers. Now I recommend this manuscript for publication in Polymers.

Reviewer 4 Report

Best wishes in your work